# Robust, independent and relevant prognostic $^{18}$F-fluorodeoxyglucose positron emission tomography radiomics features in non-small cell lung cancer: Are there any?

Tom Konert[1,2], Sarah Everitt[3,4], Matthew D. La Fontaine[2], Jeroen B. van de Kamer[2], Michael P. MacManus[3,4], Wouter V. Vogel[1,2], Jason Callahan[3], Jan-Jakob Sonke[2]*

1 Nuclear Medicine Department, Netherlands Cancer Institute, Amsterdam, The Netherlands, 2 Department of Radiation Oncology, Netherlands Cancer Institute, Amsterdam, The Netherlands, 3 Division of Radiation Oncology and Cancer Imaging, Peter MacCallum Cancer Centre, Melbourne, Australia, 4 Sir Peter MacCallum Department of Oncology, University of Melbourne, Melbourne, Australia

* j.sonke@nki.nl

**Data Availability Statement:** Data cannot be shared publicly without approval by our

## Abstract

In locally advanced lung cancer, established baseline clinical variables show limited prognostic accuracy and $^{18}$F-fluorodeoxyglucose positron emission tomography (FDG PET) radiomics features may increase accuracy for optimal treatment selection. Their robustness and added value relative to current clinical factors are unknown. Hence, we identify robust and independent PET radiomics features that may have complementary value in predicting survival endpoints. A 4D PET dataset (n = 70) was used for assessing the repeatability (Bland-Altman analysis) and independence of PET radiomics features (Spearman rank: |ρ|<0.5). Two 3D PET datasets combined (n = 252) were used for training and validation of an elastic net regularized generalized logistic regression model (GLM) based on a selection of clinical and robust independent PET radiomics features (GLM$_{all}$). The fitted model performance was externally validated (n = 40). The performance of GLM$_{all}$ (measured with area under the receiver operating characteristic curve, AUC) was highest in predicting 2-year overall survival (0.66±0.07). No significant improvement was observed for GLM$_{all}$ compared to a model containing only PET radiomics features or only clinical variables for any clinical endpoint. External validation of GLM$_{all}$ led to AUC values no higher than 0.55 for any clinical endpoint. In this study, robust independent FDG PET radiomics features did not have complementary value in predicting survival endpoints in lung cancer patients. Improving risk stratification and clinical decision making based on clinical variables and PET radiomics features has still been proven difficult in locally advanced lung cancer patients.

## Introduction

Despite the emergence of new technologies and treatment options such as tyrosine kinase inhibitors targeted towards mutations, and immune checkpoint inhibitors, the global survival

Institutional Review Board, because of institutional policy (legal restrictions). Researchers who meet the criteria for access to confidential data should request approval via IRB@nki.nl. After approval is obtained, data is available from the PET Radiomics Data Access Committee. Contact via t.konert@nki. nl (Chair of the Board).

**Funding:** One co-author, SE, received the Australian National Health and Medical Research Council (APP1003895) and Victorian Cancer Agency Grant. Sources: http://www. victoriancanceragency.org.au/ https://www.nhmrc. gov.au/ The funders had no role in study design, data collection and analysis, decision to publish, or preparation of the manuscript.

**Competing interests:** The authors have declared that no competing interests exist.

of lung cancer patients has improved only gradually in the last decades [1–4]. Locally advanced non-small cell lung cancer (NSCLC) is a highly heterogeneous disease where only modest improvements in survival have been observed, with the exception of chemoradiotherapy (CRT) patients treated with the anti-PD-L1 antibody Durvalumab whose overall and progression-free survival significantly improved compared to those receiving CRT alone [5]. New approaches are urgently needed for the selection of treatment strategies for NSCLC patients, which are currently determined mainly by TNM staging [6, 7]. In addition to TNM staging, other well-established, reproducible, independent prognostic factors are used to guide clinicians in making treatment decisions, such as Eastern Cooperative Oncology Group (ECOG) performance status [8, 9], weight loss [10], and gender [11]. Numerous other biomarkers have been investigated, although less reproducible, such as histology [12], age [13], serum blood levels [14, 15], mutation status [16], and protein expression levels [17, 18]. In locally advanced NSCLC, treatment selection based on TNM staging and other clinical variables may not be accurate enough for survival probability prediction [19, 20]. Therefore, the search for more accurate reproducible independent prognostic factors is warranted in the context of personalized medicine.

A current field of interest is the assessment of quantitative image features and its complementary value to well-established clinical prognostic models. Radiomics has been introduced as a sophisticated way to extract and mine a large number of quantitative image features, primarily using anatomical CT information [21]. The basic assumption of radiomics is that underlying tumour biology could be captured [22]. This information may actually be better characterized with functional imaging such as $^{18}$F-fluorodeoxyglucose Positron Emission Tomography (FDG PET), the gold standard in NSCLC diagnosis and staging, which is able to characterize molecular heterogeneity in lung cancer [23, 24]. It is therefore worthwhile to investigate the prognostic performance of radiomics features from functional imaging such as PET.

Basic PET radiomics features have provided clinically relevant prognostic information for NSCLC patients. Examples include standardized uptake value (SUV) based metrics like maximum, peak, and mean SUV ($SUV_{max}$, $SUV_{peak}$, and $SUV_{mean}$, respectively), metabolic tumour volume (MTV), and total lesion glycolysis (TLG) [25–32]. The more advanced PET texture features employed for quantification of tumour heterogeneity, have also been reported to be of prognostic value [33–41]. However, the variable nature of PET imaging makes it difficult to reproduce these results [42, 43].

Furthermore, PET texture features can also be subject to differences in reconstruction settings and delineation methods [44], SUV binning methods [45, 46], and feature calculation methods [47]. It is not yet clear which PET radiomics features are insensitive to all of these factors, and also to what degree.

Regardless of the issues with variability, complementary PET radiomics features should be independent from well-known prognostic SUV metrics, such as MTV and $SUV_{max}$. Some investigators reported specific PET texture features that were associated with MTV [37, 39, 47, 48, 49]. In these cases, prognostic texture features would rather act as a surrogate than as an independent variable. Such an association is also not warranted for clinical variables. Hence, the relationship of PET texture features with well-known prognostic factors has to be thoroughly studied too.

With all the confounding factors described above, in combination with the high number of possible radiomics features, it is not surprising that false discovery rates are high amongst FDG PET and CT studies on texture features [50]. Without proven, robust, and independent prognostic PET texture features, it will be challenging to move further in the field. Therefore, this study aims to investigate the repeatability of PET radiomics features, and also assesses the

relationship with well-known prognostic factors in PET, such as MTV and $SUV_{max}$. The rationale is to identify a group of radiomics features derived from pre-treatment PET imaging that are robust, independent, and prognostic, with possible additional value to current clinical prognostic variables.

## Materials and methods

### Patient data

Three NSCLC patient cohorts from the Netherlands Cancer Institute (NKI) and one from the Peter MacCallum Cancer Centre (PMCC) were included in this study to develop and validate a radiomics signature. Peter MacCallum Cancer Centre Ethics and Clinical Research Committees approval was granted and all research was performed in accordance with relevant guidelines/regulations. Patient's written, informed consent was obtained. An overview of the datasets is given in Table 1. Patients were excluded if the primary tumour was smaller than 10 cc or if the patient had stage IV NSCLC at baseline. To detect brain metastases at baseline, the NKI patients were scanned with MR imaging and the PMCC performed FLT baseline scans before treatment.

The repeatability and independence of PET radiomics features was assessed using a 4D PET/CT dataset (4D PET lung) consisting of 70 stage III NSCLC patients. No clinical data was collected for these patients. The second cohort (NKI lung 1) contained 228 patients treated with concurrent chemoradiotherapy (CCRT) for stage IA-IIIC NSCLC in the NKI between 2007 and 2011 as described earlier [51]. The third cohort, also from the NKI (NKI lung 2), consisted of 24 patients with stage IIB-IIIC NSCLC treated between 2013 and 2016, similar as NKI lung 1. The fourth cohort was from the PMCC (PMCC lung 1) and involved 40 stage IB-IIIC NSCLC patients treated with CCRT as previously reported [32].

### Clinical endpoints for prognostic model

The primary endpoint used for the prognostic model was two-year overall survival (2-year OS). Overall survival was defined as the time between the start of treatment and date of death. In addition, two-year progression-free survival (2-year PFS), one-year PFS (1-year PFS), one-year local recurrence-free survival (1-year LRS), and one-year distant metastases-free survival (1-year DMS) were also studied. Progression was defined as growth of tumour cells in the primary tumour or involved lymph nodes, or metastases to other organs, or death. LRS was defined as progression in the primary tumour and/or involved lymph nodes as assessed on follow-up scans. DMS was described according to the 8[th] edition of the TNM classification for NSCLC [52] as evaluated on follow-up scans.

### Data acquisition and image reconstruction

Patients from the NKI lung 1 and 2 dataset both underwent a whole-body FDG PET/CT using a Gemini TF or Gemini TF Big Bore scanner (Philips Medical Systems, Cleveland, OH). The reconstruction voxel size of the PET data was $4 \times 4 \times 4$ mm[3]. Patients fasted for at least 8 h to ensure low levels of serum glucose. Patients with a Body Mass Index (BMI)$\leq$28 were intravenously injected with 190 MBq [18]F-FDG, or 240 MBq in case of a BMI$>$28. Patients were scanned 60 minutes after injection of [18]F-FDG. The acquisition time of the PET/CT scanner was 2 minutes per bed position.

In the PMCC lung 1 cohort, whole-body FDG PET/CT scans were acquired on a GE STE (GE Medical Systems, Milwaukee, WI) or Biograph (Siemens Medical Solutions, Erlangen, Germany) scanner. The reconstructed voxel size of the PET data was $4.3 \times 4.3 \times 3.3$ mm[3] for

**Table 1. Overview of the four patient cohorts used in the study.** Unless otherwise stated, values represent the median with the range in parentheses. MTV$_{2.5}$ = metabolic tumour volume obtained using a SUV threshold of 2.5, MTV$_{40}$ = metabolic tumour volume obtained using a threshold of 40% of the maximum intensity, SUV$_{max}$ = maximum SUV uptake, OS = overall survival, PFS = progression-free survival, LRS = local recurrence-free survival, DMS = distant metastases-free survival. Nos = not otherwise specified.

| | 4D PET lung | NKI lung 1 | NKI lung 2 | PMCC lung 1 |
|---|---|---|---|---|
| No. of patients | 70 | 228 | 24 | 40 |
| Age (year) | n/a | 64 (36–87) | 63 (39–82) | 68 (53–86) |
| Gender | n/a | | | |
| Male | | 142 | 13 | 29 |
| Female | | 86 | 11 | 11 |
| Disease stage | n/a | IA-IIIC | IIB-IIIC | IB-IIIC |
| IA | | 1 | 0 | 0 |
| IB | | 4 | 0 | 1 |
| IIA | | 0 | 0 | 3 |
| IIB | | 16 | 2 | 5 |
| IIIA | | 102 | 7 | 15 |
| IIIB | | 82 | 13 | 12 |
| IIIC | | 23 | 2 | 4 |
| T stage | n/a | | | |
| 1 | | 6 | 1 | 5 |
| 2 | | 78 | 4 | 24 |
| 3 | | 63 | 7 | 9 |
| 4 | | 81 | 12 | 2 |
| N stage | n/a | | | |
| 0 | | 42 | 4 | 5 |
| 1 | | 20 | 2 | 5 |
| 2 | | 130 | 13 | 17 |
| 3 | | 36 | 5 | 13 |
| Histology | n/a | | | |
| Adeno | | 80 | 12 | 16 |
| Squamous cell | | 83 | 7 | 13 |
| Large cell | | 8 | 1 | 5 |
| Nos or other | | 57 | 4 | 6 |
| GTV (cc) | n/a | 118 (15–906) | 84 (10–351) | 49 (12–544) |
| MTV$_{2.5}$ (cc) | 62 cc (10–545) | 72 (10–693) | 91 (11–337) | 51 (8–478) |
| MTV$_{40}$ (cc) | 27 cc (4–169) | 31 (3–394) | 34 (5–289) | 31 (4–378) |
| SUV$_{max}$ | 11.5 (4.3–55.1) | 14.6 (5.9–44.8) | 15.7 (6.9–28.3) | 16.5 (6.3–33.2) |
| Median follow-up time (months) | n/a | 17 | 22 | 24 |
| 2-year OS | n/a | 40% | 46% | 54% |
| 2-year PFS | n/a | 29% | 21% | 20% |
| 1-year PFS | n/a | 50% | 54% | 35% |
| 1-year LRS | n/a | 58% | 71% | 47% |
| 1-year DMS | n/a | 54% | 54% | 45% |

the GE STE scanner, and 4.1 × 4.1 × 3.0 mm$^3$ for the Siemens Biograph scanner. Patients fasted for more than 6 hours before $^{18}$F-FDG scans. Patients were intravenously injected with 4.2 MBq/kg $^{18}$F-FDG. Baseline emission scans were initiated 60 minutes after injection. The acquisition time of the PET/CT scanner was 3 minutes per bed position.

For the 4D PET lung dataset, scans were acquired on a Gemini TF scanner (Philips Medical Systems, Cleveland, OH). The reconstruction voxel size of the PET data was 4 × 4 × 4 mm$^3$.

The 4D PET/CT data were reconstructed in 10 phases, and the attenuation in each frame of the 4D PET data was corrected with the corresponding 4D CT frame. The acquisition time of the 4D PET was kept the same as that used for 3D PET [52].

## Mid-position scans from 4D PET lung dataset for repeatability testing

The 4D PET/CT data were reconstructed in 10 phases, and from these phases two new mid-position scans were derived [53]. The first mid-position scan was created from the even phases (0, 2, 4, 6, and 8) and is named 'Mid-P even', and the odd phases (1, 3, 5, 7, and 9) were used to create the second mid-position scan 'Mid-P odd'. The even and odd number of frames were selected to keep the amount of tumour motion balanced in both scans. Fig 1 gives an overview of the workflow.

The source of variability was different in these two mid-position scans compared to a test–retest setting, since the biological tumour variability has been eliminated. In this case, the variability was mostly caused by minor differences in noise-levels and tumour motion, hence robust quantitative features should not differ substantially in outcome.

## Tumour segmentation

For each patient in the NKI lung 1, NKI lung 2, and PMCC lung 1 cohort, a volume-of-interest (VOI) enveloping the primary tumour was manually drawn by radiation oncologists using information from both PET and CT imaging. From this VOI, the MTV was auto-segmented

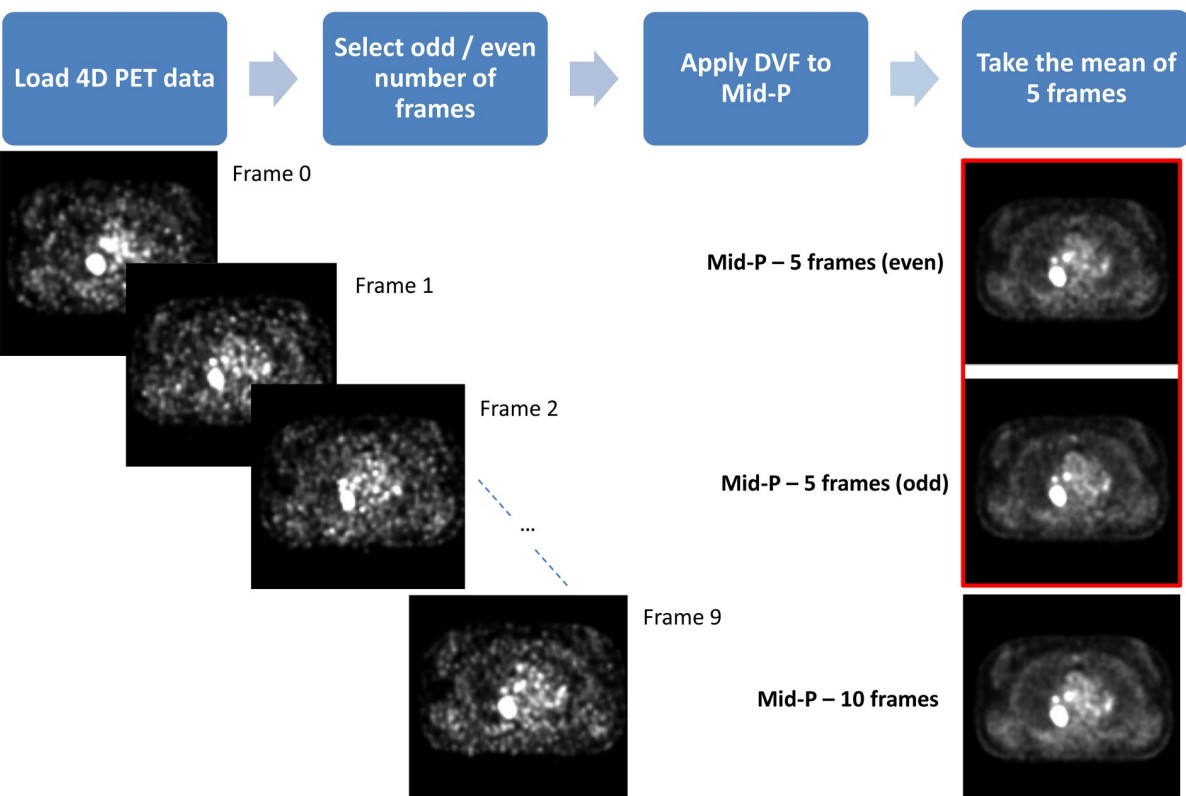

**Fig 1. Workflow of the PET mid-position scans.** A 4D PET scan was loaded for each patient consisting of 10 frames, where the odd or even number of frames were selected. A 4D deformation vector field (DVF) was applied to these frames to deform them to the mid-position. Lastly, the mean of the 5 deformed frames was calculated to obtain the PET mid-position scan. For comparison, the PET mid-position scan obtained from 10 frames has been included in the image too. Mid-P = PET mid-position scan.

on the FDG PET scan. Two auto-segmentation methods were applied: a metabolic tumour region delineation that included all voxel intensities above 2.5 ($SUV_{2.5}$), and a high intensity delineation that included all voxel intensities that were at least 40% of the $SUV_{max}$ ($SUV_{40}$). Auto-segmentation was performed with in-house developed software named Match42 (version 1.0.0) using a Python plug-in. The metabolic tumour volume obtained from $SUV_{2.5}$ and $SUV_{40}$ were named $MTV_{2.5}$ and $MTV_{40}$, respectively. In the 4D PET lung dataset, a VOI was manually drawn around the primary tumour in one PET mid-position scan, and copied to the second PET mid-position scan. The auto-segmentation was performed on both PET mid-position scans independently.

## PET radiomics features

The Pyradiomics toolkit was used for radiomics feature extraction [54]. With this toolkit a total of 105 features were available for feature calculations. These were divided into 18 first-order features, 13 shape features (including metabolic tumour volume), and 74 texture features describing the spatial distribution of voxel intensities. The texture features were derived from the gray level co-occurrence matrix (GLCM; 23 features) [55], gray level run-length matrix (GLRLM; 16 features) [56], gray level size-zone matrix (GLSZM; 16 features) [57], gray level dependence matrix (GLDM; 14 features) [58], and neighbourhood gray tone difference matrix (NGTDM; 5 features) [59]. The mathematical definitions of these features were in compliance with feature definitions as described by the Imaging Biomarker Standardization Initiative (IBSI) [60].

## SUV discretization and matrix calculation

Before texture features were extracted, pre-processing steps were required in the form of SUV binning and matrix definition. SUV discretization is an intensity-resampling step, before building the texture matrices on which texture features rely. SUV discretization or binning was applied with the fixed bin count method (e.g. 64 bins) and an alternative method using a fixed bin width (e.g. 0.25 SUV). All texture features were calculated from a single matrix taking into account all 13 directions simultaneously. A more detailed description on SUV binning and matrix calculation can be found in S1 File, respectively.

## Repeatability

The repeatability assessment was performed within the same patient comparing two different PET mid-position scans. For each patient, the PET mid-position scan obtained from the even numbered frames (Mid-P even) was compared with the PET mid-position scan from the odd numbered frames (Mid-P odd). This resulted in four comparisons: 2 SUV binning methods and 2 thresholding methods were applied.

The repeatability of each PET radiomics feature was assessed with the Coefficient of Repeatability (CR) [61]. See S1 File for more details. The CR was reported as a percentage: $100\% \times \frac{CR}{mean}$, where *mean* is the average of the PET radiomics feature value within the patient cohort. The threshold for poor repeatability was set to a value of 30%, corresponding to PET Response Criteria in Solid Tumours (PERCIST) [62].

## Independence testing

To determine whether the features were correlated with the two commonly reported prognostic PET features MTV and $SUV_{max}$, the Spearman's rank correlation coefficient ($\rho$) was calculated on one of the Mid-P scans, using the same set-up as for the repeatability testing. PET

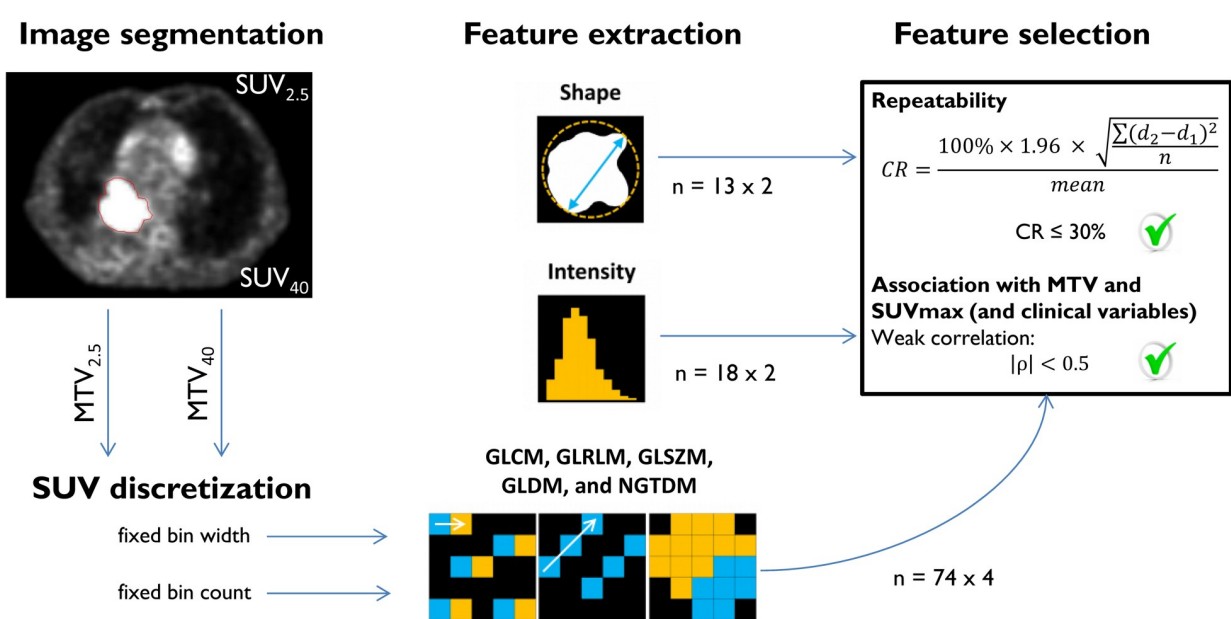

**Fig 2. Radiomics feature selection workflow: From PET image segmentation to selected features.** Features from $MTV_{2.5}$ and $MTV_{40}$ were seen as a separate set of features, doubling the amount of features in the analysis. This also counts for features calculated with fixed bin width and fixed bin count, except for most intensity and shape features that were not affected by SUV discretization. An exception was observed for first-order features Uniformity and Entropy. A total of 360 PET radiomics features were entered into the analysis, including $SUV_{max}$, $MTV_{2.5}$, and $MTV_{40}$. PET radiomics features were selected for further analysis when two criteria were met: high repeatability and low association with MTV and $SUV_{max}$. $SUV_{2.5}$ = SUV threshold of 2.5; $SUV_{40}$ = SUV threshold of 40% of maximum SUV; $MTV_{2.5}$ = metabolic tumour volume obtained from use of $SUV_{2.5}$; $MTV_{40}$ = metabolic tumour volume obtained from use of $SUV_{40}$. GLCM = gray level co-occurrence matrix; GLRLM = gray level run-length matrix; GLSZM = gray level size-zone matrix; GLDM = gray level dependence matrix; NGTDM = neighbourhood gray tone difference matrix; CR = coefficient of repeatability.

radiomics features that had a $|\rho| \geq 0.5$ were considered to have a correlation with MTV or $SUV_{max}$, and were discarded from further analysis. The choice of $|\rho| < 0.5$ as limit for independent features was validated with the 'elbow method' using hierarchical clustering [63].

An overview of the radiomics workflow and feature selection procedure is given in Fig 2.

## Model training

An elastic net regularized generalized logistic regression model (GLM) was built with PET radiomics features derived from pre-treatment PET imaging ($GLM_{rad}$). To increase the sample size in the training and test sets, for the purpose of building a GLM, NKI lung 1 and lung 2 were combined. In this study, 80% of the NKI data was used for training the model, and 20% for validation. Different ratios of training/validation were also tested, but were not reported as there was no major differences seen in the results. Elastic net regression analysis using the R package 'glmnet' was performed on the training set [64]. With 20-fold cross validation (CV), the most optimal fitted $GLM_{rad}$ with minimal CV error was determined and selected for model validation.

## Model validation

To validate the fitted model of the training set, the area under the receiver operating characteristic curve (AUC) was calculated between the predicted outcome and the observed outcome in the validation set. To reduce randomness introduced by selecting a random subset of the complete data for training and validation, the procedure for model training and validation was

repeated 100 times. This yields a better estimate of the true validation set performance by randomly simulating many scenarios with varying training and validation set compositions [65]. From the 100-times-repeated training/validation procedure, results were averaged, and the best performing $GLM_{rad}$ was externally validated for each clinical endpoint on PMCC lung 1.

During 100-times-repeated training/validation procedure, per iteration, the fitted model was stored to keep track of the PET radiomics features that were selected by elastic net in the fitted model [66]. PET radiomics features and clinical variables were ranked based on the frequency of inclusion in the fitted model.

## Model comparison

Clinical variables such as PET/CT-based GTV, TNM staging, histology, gender, and age were also introduced into the radiomics signature to create a prognostic model containing PET radiomics features and clinical variables ($GLM_{all}$). In addition, a model based on only the clinical variables was calculated using elastic net regression ($GLM_{clin}$). To assess the complementary value of PET radiomics features with clinical variables, the mean AUC was calculated from 100 iterations for each model and compared. The Mann Whitney U Test was used to assess any significant differences between the predictive performance of $GLM_{all}$, $GLM_{clin}$, and $GLM_{rad}$, and p-values below 0.05 were seen as significant.

# Results

## Repeatability

Results of the repeatability test were based on the 4D PET lung dataset and an overview of notable PET radiomics features and their corresponding CR is given in Table 2. All first-order features were repeatable when extracted from $MTV_{2.5}$ irrespective of SUV binning method. In contrast, 13 out of 18 first-order features were repeatable when extracted from $MTV_{40}$. Furthermore, around 50 texture features were repeatable when extracted from $MTV_{2.5}$ regardless of SUV discretization method, versus 28 repeatable texture features extracted from $MTV_{40}$. With regards to shape features, only $MTV_{40}$ was not repeatable.

Amongst the four comparisons, 211 out of 360 PET radiomics features were repeatable. An overview of all PET radiomics features and their corresponding CR is given in S1 File. The impact of large delineation inaccuracies on repeatability was studied between contours generated by the two different SUV thresholds, though only reported as supplementary data (S1 File).

## Relationship of PET radiomics features with MTV and $SUV_{max}$

The Spearman's Rank correlation coefficient was calculated to assess the relationship of 211 repeatable PET radiomics features with MTV and $SUV_{max}$. Four assessments were performed in total on one of the mid-position scans, with groups consisting of a combination of either one of the SUV binning methods and one of the tumour volumes ($MTV_{2.5}$ or $MTV_{40}$). Not all repeatable PET radiomics features were found to be independent from MTV and $SUV_{max}$. From the first-order features, only Kurtosis and Skewness extracted from $MTV_{2.5}$ were independent from MTV and $SUV_{max}$. There were no independent repeatable first-order features for $MTV_{40}$. Regarding the fixed bin count method, 17 out of 50 texture features extracted from $MTV_{2.5}$ were not strongly associated with MTV and $SUV_{max}$. This also counted for 5 texture features extracted from $MTV_{40}$. With regards to the fixed bin width method, there were no texture features independent from either $SUV_{max}$ or MTV. Elongation, Flatness, and Sphericity were the only independent shape features when extracted from $MTV_{2.5}$, though only

**Table 2. An overview of categorized notable PET radiomics features that are commonly reported in literature with their coefficient of repeatability (CR, %).** The asterisk (*) represents features that were repeatable in all four different settings. Per category, the total number of PET radiomics features that met the study repeatability criterion is added.

| CR (%) | Fixed bin width | | Fixed bin count | |
|---|---|---|---|---|
| Notable features | MTV$_{2.5}$ | MTV$_{40}$ | MTV$_{2.5}$ | MTV$_{40}$ |
| **First-order features** | **18/18** | **13/18** | **18/18** | **13/18** |
| Entropy* | 3.4 | 5.5 | 3.8 | 6.0 |
| Kurtosis | 26.8 | 34.7 | 26.8 | 34.8 |
| Skewness | 23.1 | 50.4 | 23.1 | 51.3 |
| SUVmax* | 13.2 | 13.2 | 13.2 | 13.2 |
| SUVmean* | 6.0 | 12.9 | 6.0 | 12.7 |
| Uniformity | 17.9 | 41.9 | 21.1 | 37.0 |
| **Texture features** | **49/74** | **28/74** | **50/74** | **28/74** |
| GLCM Contrast* | 23.2 | 28.1 | 28.8 | 29.9 |
| GLCM Correlation* | 2.6 | 11.9 | 2.7 | 11.2 |
| GLCM DifferenceAverage* | 9.9 | 13.1 | 14.1 | 17.2 |
| GLCM JointEntropy* | 2.8 | 4.5 | 3.3 | 5.7 |
| GLCM SumEntropy* | 2.7 | 4.5 | 2.8 | 4.2 |
| GLRLM GrayLevelNonUniformity | 13.7 | 59.1 | 18.7 | 55.4 |
| NGTDM Busyness | 75.5 | 91.3 | 33.0 | 81.9 |
| NGTDM Coarseness | 12.0 | 41.5 | 16.8 | 35.2 |
| NGTDM Contrast | 23.3 | 68.5 | 31.9 | 64.9 |
| **Shape features** | **13/13** | **12/13** | **13/13** | **12/13** |
| Elongation* | 4.7 | 10.8 | 4.7 | 10.7 |
| Flatness* | 7.1 | 15.0 | 7.1 | 13.7 |
| MetabolicTumourVolume | 5.9 | 45.5 | 5.9 | 45.3 |
| Sphericity* | 3.2 | 8.9 | 3.2 | 8.7 |

Elongation and Flatness remained independent for MTV$_{40}$. A complete overview of independence testing for all PET radiomics features is given in S1 File.

An overview of correlations amongst the selected robust independent PET radiomics features and clinical variables is given in Fig 3. More details on robust and independent PET radiomics features can be viewed in S1 File. The robust independent PET radiomics features did not show any strong correlation with the other clinical variables, such as age, ECOG PS, gender, histology, and TNM stage. However, there were associations present amongst the PET texture features.

## Building the radiomics signature

Based on the feature selection criteria, 31 PET radiomics features were selected for the next steps (see Fig 3). Three elastic net regularized GLMs were built per endpoint: GLM$_{rad}$, GLM$_{clin}$, and GLM$_{all}$. Results of the model performances are shown in Fig 4, showing that GLM$_{rad}$ does not significantly outperform GLM$_{clin}$ for any clinical endpoints. The GLM$_{clin}$ has a significantly better predictive performance compared to GLM$_{rad}$ in 2-year OS (p<0.0001), and in 1-year LRS (p<0.001). GLM$_{all}$ did not show a significantly better performance to both GLM$_{rad}$ and GLM$_{clin}$ simultaneously in any endpoint. External validation of GLM$_{all}$ led to AUC values ranging from 0.51 to 0.59 for any clinical endpoint. When GLM$_{clin}$ was externally validated, the highest predictive performance was 0.60 for 2 year OS. For GLM$_{rad}$, the highest predictive performance was 0.71 for 2-year PFS.

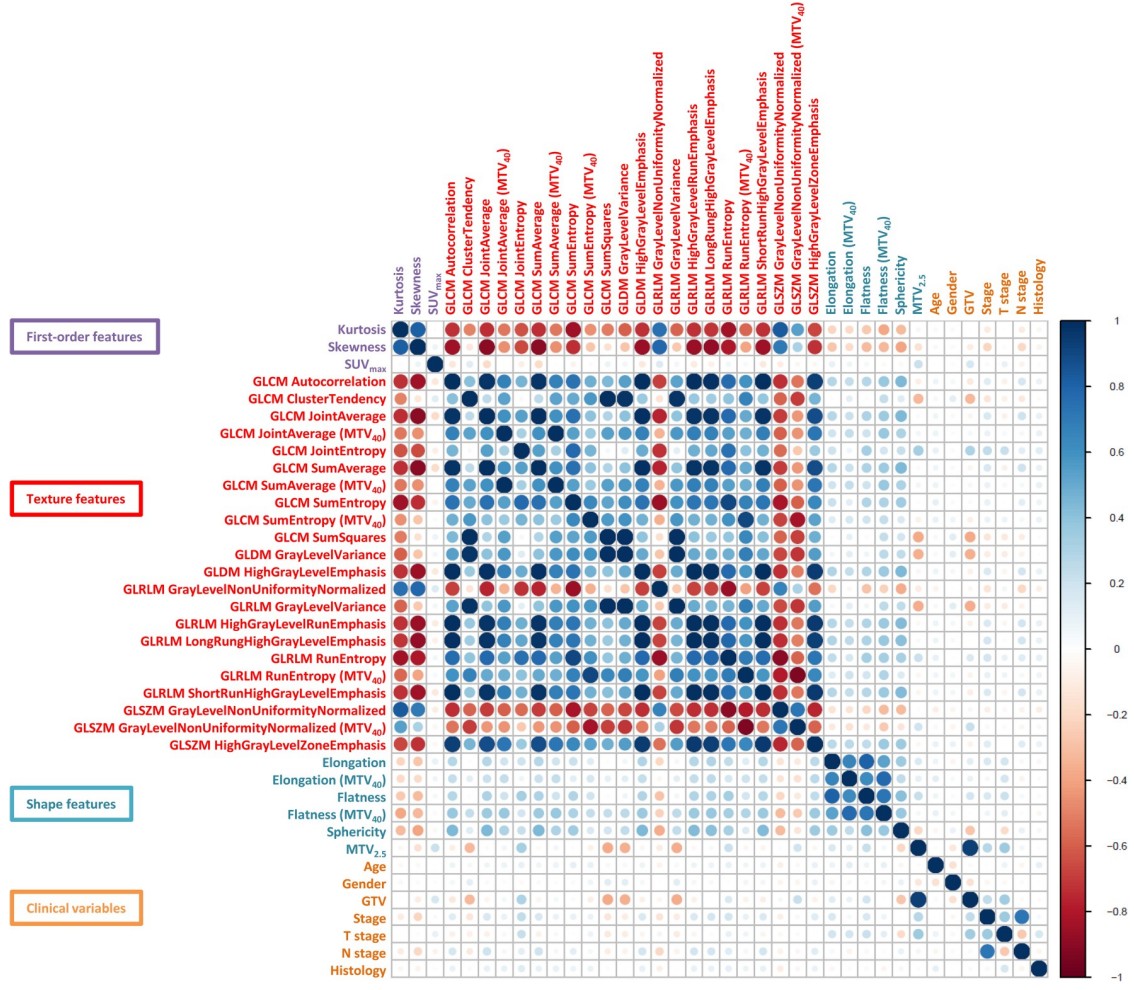

**Fig 3. Correlation coefficients of the robust independent PET radiomics features and clinical variables.** Positive correlation coefficients are displayed in blue and negative correlation coefficients in red color. Color intensity and the size of the circle are proportional to the correlation coefficients. A distinction was made between features calculated from $MTV_{2.5}$ and $MTV_{40}$.

## Promising features

Table 3 shows selected features for each fitted GLM, and how frequent these features were chosen in the fitted model over 100 iterations. The feature shape Sphericity was present in 100% of the iterations for 2-year OS. From the 100 repetitions, GLCM ClusterTendency was selected in more than 95% for predicting 1-year PFS and 1-year DMS. Clinical variables such as age and GTV were prominent in predicting 2-year OS and 1-year LRS, next to shape Sphericity. As can be seen in Table 3, age, shape Sphericity, and GLCM ClusterTendency are present amongst the most selected features for all clinical endpoints.

## Discussion

The rationale of this study was to identify a group of FDG PET radiomics features for NSCLC patients that are robust, independent, prognostic, and complementary to well-established clinical variables. We found PET radiomics features that met the study criteria of robustness and independence, and that also exhibited prognostic value. However, results demonstrated that PET radiomics features are not complementary to clinical variables for predicting clinical

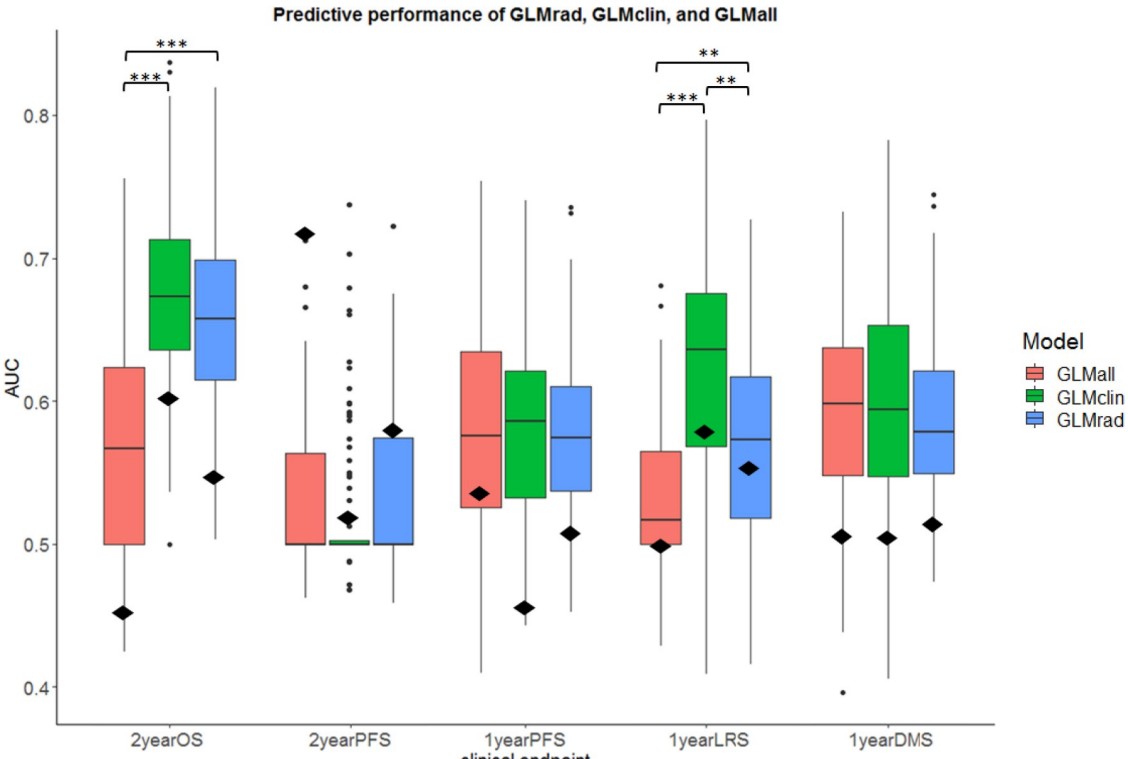

**Fig 4. Model performance for the PET radiomics model (GLM_rad), the model containing clinical variables (GLM_clin), and a combination of radiomics and clinical variables (GLM_all).** The median AUC values from 100-times-repeated training/validation are depicted per model, per clinical endpoint. The lower and upper hinges correspond to the 25th and 75th percentiles. The whiskers depict the 1.5*IQR from the lower and upper hinge. Data beyond the end of the whiskers are shown as outlier points. AUC values corresponding to the external validation set are shown as a black diamond. Significance levels, **p<0.001, ***p<0.0001.

endpoints in NSCLC patients that were treated with CCRT. This indicates that clinical variables provide more prognostic information than robust independent PET radiomics features, and that the prognostic value in PET radiomics features is minimal. This study did take into account shortcomings of other studies on PET radiomics features [50] with the use of a feature selection method that reduces overfitting and external validation of results.

Feature selection based on the repeatability of PET radiomics features was feasible with the use of different phases from 4D PET imaging, in the absence of test-retest data. Larue et al. showed that in 4D CT, the majority of the features have a high agreement between radiomics feature stability based on 4D CT and test–retest data in lung cancer [67]. It was therefore hypothesized that 4D PET scans could also be used for repeatability testing. To determine robust PET radiomics features, a CR of 30% was chosen as limit for repeatability, based on PERCIST. However, a limitation of using 4D PET for repeatability testing is the absence of biological tumour variability, and PERCIST takes this variability into account. Hence, the use of a 30%-limit could be seen as too tolerant, and 15%, as commonly used in phantom studies, could be more appropriate. Even under these stricter circumstances, 12 first-order features, 24 out of 74 texture features, and all shape features would still meet that criterion as can be seen in S1 File. Besides that, the most prominent PET radiomics features in the fitted GLMs were $SUV_{max}$ (CR = 13.2%), shape Sphericity (CR = 3.2%), GLCM ClusterTendency (CR = 21.9%), GLRLM GrayLevelNonUniformityNormalized (CR = 18.4%), and $MTV_{2.5}$ (CR = 5.9%) as seen in Table 3. This shows that repeatable PET radiomics features with a CR>15% are also

**Table 3. The most selected features in the model by elastic net, ranked by the number of times selected in the generalized linear model.** Only the top 10 most selected PET radiomics are shown. The features written in italic bold are present in all endpoints.

| Endpoint | GLM$_{all}$ selected features by elastic net | Frequency |
|---|---|---|
| **2-year OS** | ***Age*** | 100 |
| | GTV | 100 |
| | ***Shape_Sphericity*** | 100 |
| | MTV$_{2.5}$ | 78 |
| | ***glcm_ClusterTendency*** | 56 |
| | SUV$_{max}$ | 39 |
| | Gender | 34 |
| | glcm_JointEntropy | 34 |
| | glrlm_GrayLevelNonUniformityNormalized | 33 |
| | glrlm_GrayLevelVariance | 29 |
| **2-year PFS** | ***Age*** | 50 |
| | SUV$_{max}$ | 50 |
| | glrlm_GrayLevelNonUniformityNormalized | 49 |
| | ***shape_Sphericity*** | 47 |
| | Histology | 42 |
| | MTV$_{2.5}$ | 38 |
| | ***glcm_ClusterTendency*** | 30 |
| | N_status | 28 |
| | T_status | 25 |
| | shape_Elongation_MTV40 | 20 |
| **1-year PFS** | GTV | 99 |
| | ***glcm_ClusterTendency*** | 95 |
| | ***shape_Sphericity*** | 76 |
| | ***Age*** | 63 |
| | T_status | 49 |
| | MTV$_{2.5}$ | 40 |
| | glcm_SumEntropy_MTV40 | 39 |
| | shape_Elongation | 37 |
| | SUV$_{max}$ | 31 |
| | Histology | 29 |
| **1-year LRS** | GTV | 83 |
| | ***Age*** | 82 |
| | ***glcm_ClusterTendency*** | 65 |
| | glcm_SumEntropy_MTV40 | 63 |
| | ***shape_Sphericity*** | 57 |
| | Gender | 48 |
| | gldm_GrayLevelVariance | 47 |
| | N_status | 27 |
| | Stage | 25 |
| | glrlm_GrayLevelVariance | 24 |
| **1-year DMS** | GTV | 99 |
| | ***glcm_ClusterTendency*** | 96 |
| | ***shape_Sphericity*** | 57 |
| | MTV$_{2.5}$ | 52 |
| | Histology | 34 |
| | T_status | 33 |
| | ***Age*** | 32 |
| | glcm_SumEntropy_MTV40 | 32 |
| | shape_Elongation | 29 |
| | SUV$_{max}$ | 23 |

frequently present in the fitted models. Even though there is literature reporting on stability of PET radiomics features in a test-retest setting [45, 46], there is no objective limit for the level of repeatability for each PET radiomics feature. Determining such an objective limit is only relevant if the studied PET radiomics feature contains clinically useful information. Hence, in the absence of an objective limit for each PET radiomics feature, the 30%-limit of PERCIST was applied to all.

It was observed that the repeatability for features from MTV2.5 is better compared to MTV40 and this is due to two important factors:

1. From the 13 shape features, only MTV40 had a CR>30% when comparing the MTV40 between two mid-position scans. This variance, of course, has already a great impact on PET radiomics features calculated from MTV40 as it is known that differences in delineation have an impact on feature outcome [44].

2. Radiomics features are calculated on matrices which dimensions are dependent on the SUV range. With MTV2.5 matrix dimensions are more standardized than MTV40, which is dependent on the maximum SUV (CR = 13.2%).

In this case, the use of MTV2.5 for GTV delineation may be advised over MTV40 in PET radiomics analysis.

Another step of the feature selection procedure was to assess the independence of PET radiomics features, to identify possible prognostic features that could complement basic SUV metrics and volumetric features. In this context, changes in PET radiomics features would be independent from changes in basic SUV metrics and volume, increasing their utility in longitudinal studies. Therefore, the use of a fixed bin width for SUV binning should be avoided as this method resulted in PET radiomics features that were all strongly correlated to either maximum SUV or MTV. While the choice of $|\rho|<0.5$ for independence testing may seem arbitrary, a $|\rho|<0.7$ was also studied and did not improve results (see S1 File for more details). Independence testing had the most impact in the feature pre-selection procedure as it resulted in a substantial decrease of PET radiomics features. Unfortunately, results demonstrated that independence testing could not guarantee that remaining robust independent PET radiomics features exhibited complementary value next to clinical variables. Even so, we strongly advise assessing the relationship of radiomics features with current established prognostic factors in any study considering PET radiomics features for prognostication as this is the first important step in showing their potential added value in the clinic.

A final selection of features in the GLM was performed by elastic net regression, robust to collinearity amongst features [66]. More feature selection/classification methods exist [68], though comparing multiple methods was beyond the scope of this study. However, in literature, elastic net regression yielded one of the highest discriminative performances in chemoradiotherapy outcome prediction in 12 patient datasets containing in total 1053 lung cancer patients [65]. Interestingly, elastic net regression could also be used as a standalone feature selection method. A comparison of the feature selection method based on repeatability, independence, and elastic net regression (GLM$_{all}$), and a method using only elastic net regression (GLM$_{elnet}$) was performed, see S1 File. Pre-selection of PET radiomics features is worthwhile, because the number of PET radiomics features in GLM$_{elnet}$ was often high (>20 features) and many were highly correlated to volume or SUV$_{max}$. In contrast, the average number of features in GLM$_{all}$ was 9. Even so, it was observed that elastic net tends to keep all of the correlated and presumably prognostic features in the fitted model or shrinks all to zero, whereby increasing the number of (correlated) features resulted in a decrease of the predictive performance. This decrease of predictive performance seen in the validation set suggests that overfitting, although

reduced, may still be present. This shows the value of dimensionality reduction in order to optimize predictive performance in rather small sample sizes.

The predictive performance of PET texture features in NSCLC has been studied widely, but clear evidence that PET texture features are complementary to clinical variables is lacking [69]. This study has extensively studied PET texture features and did not find any evidence for added value next to current clinical variables. S1 File provides a complete overview of all assessed model performances, including additional investigations with TLG. In literature, typically, only one or two PET texture features have been significantly associated with predicting various survival endpoints [39–41, 47, 70–72]. However, of all the prognostic PET texture features from those studies, such as GLCM Joint Entropy, Correlation, Contrast, Dissimilarity (or Difference Average), NGTDM Coarseness, Busyness, and Contrast, only GLCM Joint Entropy was both repeatable and independent from $SUV_{max}$ or volume in our dataset. In this study, GLCM Joint Entropy was selected 34 times out of 100 by elastic net regression for predicting 2-year OS, and its value in overall survival was also previously shown [47]. Nonetheless, in our study the average predictive performance for $GLM_{all}$ in all clinical endpoints ranged from 0.50 to 0.66. For comparison, other studies predicting outcome with both PET radiomics and clinical variables in NSCLC found predictive performances of 0.63 for predicting OS [41], 0.72 for local recurrence [71], and 0.71 for distant metastases [72]. Even with those results, neglecting any limitations of those studies, there is still no strong evidence that PET texture features exhibit complementary information.

Results from the external validation demonstrated even lower AUC values in most cases than the internal validation set. Besides the limitation of the use of a small external dataset, differences were observed between institutes regarding patients, treatment, and image acquisition and reconstruction settings, that also can influence outcome [44, 73], and could have resulted in poor generalizability. To overcome the issue of poor generalizability, a prognostic model should be trained on a combination of well-balanced patient cohorts from multiple institutes, and PET acquisition and reconstruction protocols should be harmonized across centers in multi-centre studies. Alternatively, a post-reconstruction harmonization method proposed by Orlhac et al. may also aid in removing the multicenter effect for textural features and SUV [74].

Furthermore, limitations of this paper include the relatively small sample size for machine learning methods that could have affected the predictive performance [75], and the impact of tumour motion on PET radiomics features, especially in lower lobe tumours [76]. Although Grootjans et al. showed that there are specific PET radiomics features whose prognostic accuracy was not affected by respiratory motion and varying noise-levels [29].

To overcome the limitations of this study, and to be certain that there is no complimentary information in PET radiomics features, future studies need to set up large scale multi-centre cohorts to allow for multiple independent validation datasets. To further improve predictive performance, studies could investigate elastic net-Cox proportional hazard models [77], non-linear relationships by applying data transformation on PET radiomics features [21, 78, 79], or assess computer engineered features with neural networks or deep learning networks [80, 81]. Currently, deep learning is under investigation for use in lung nodule detection, tumour segmentation, and tumour classification with histopathology images [82]. Its use in medical image analysis is increasing as algorithms become more sophisticated and more data becomes available, which might lead to new insights in survival prediction. A step further would be to combine radiomics features from multimodal imaging such as PET, CT and MRI [83, 84], where the combination of anatomical and biological features may of added value for providing a personalized treatment strategy.

## Conclusion

In this study, robust independent PET radiomics features, identified with 4D PET imaging, did not have complementary value in predicting overall survival and progression-free survival in NSCLC patients treated with concurrent chemoradiotherapy. Improving risk stratification and clinical decision making based on clinical variables and PET radiomics features has still been proven difficult in locally advanced lung cancer patients. New approaches should be investigated in large scale multi-centre studies to deal with current challenges in the field of radiomics before translation to the clinic becomes realistic.

## Supporting information

**S1 File.**
(DOCX)

## Acknowledgments

The authors would like to thank Simon van Kranen and Jonas Teuwen for helping out with programming, Michel van den Heuvel for providing NKI lung 1 dataset, Natascha Bruin for providing delineations and clinical data for the NKI lung 2 dataset, and Erik Vegt, Rod Hicks, Nick Hardcastle, David Ball, and Tomas Kron for scientific editing of the manuscript.

## Author Contributions

**Conceptualization:** Tom Konert, Sarah Everitt, Matthew D. La Fontaine, Jeroen B. van de Kamer, Wouter V. Vogel, Jan-Jakob Sonke.

**Data curation:** Tom Konert, Sarah Everitt, Jeroen B. van de Kamer, Jason Callahan.

**Formal analysis:** Tom Konert.

**Investigation:** Tom Konert.

**Methodology:** Tom Konert, Jeroen B. van de Kamer, Jan-Jakob Sonke.

**Project administration:** Tom Konert.

**Resources:** Tom Konert.

**Software:** Tom Konert, Matthew D. La Fontaine.

**Supervision:** Matthew D. La Fontaine, Wouter V. Vogel, Jan-Jakob Sonke.

**Validation:** Tom Konert, Jan-Jakob Sonke.

**Visualization:** Tom Konert.

**Writing – original draft:** Tom Konert.

**Writing – review & editing:** Sarah Everitt, Matthew D. La Fontaine, Jeroen B. van de Kamer, Michael P. MacManus, Wouter V. Vogel, Jason Callahan, Jan-Jakob Sonke.

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
