## [Decision Letter · Decision Letter 0]

5 Dec 2019

PONE-D-19-30593

Robust, independent and relevant prognostic 18F-fluorodeoxyglucose positron emission tomography radiomics features in non-small cell lung cancer: are there any?

PLOS ONE

Dear Mr Konert,

Thank you for submitting your manuscript to PLOS ONE. After careful consideration, we feel that it has merit but does not fully meet PLOS ONE’s publication criteria as it currently stands. Therefore, we invite you to submit a revised version of the manuscript that addresses the points raised during the review process.

We would appreciate receiving your revised manuscript by Jan 19 2020 11:59PM. To enhance the reproducibility of your results, we recommend that if applicable you deposit your laboratory protocols in protocols.io, where a protocol can be assigned its own identifier (DOI) such that it can be cited independently in the future. For instructions see: http://journals.plos.org/plosone/s/submission-guidelines#loc-laboratory-protocols

We look forward to receiving your revised manuscript.

Kind regards,

Domenico Albano

Academic Editor

PLOS ONE

Journal Requirements:

We noticed you have some minor occurrence(s) of overlapping text with the following previous publication(s), which needs to be addressed:

2. In your revision ensure you cite all your sources (including your own works), and quote or rephrase any duplicated text outside the Methods section. Further consideration is dependent on these concerns being addressed.

a) If there are ethical or legal restrictions on sharing a de-identified data set, please explain them in detail (e.g., data contain potentially identifying or sensitive patient information) and who has imposed them (e.g., an ethics committee). Please also provide contact information for a data access committee, ethics committee, or other institutional body to which data requests may be sent.b) If there are no restrictions, please upload the minimal anonymized data set necessary to replicate your study findings as either Supporting Information files or to a stable, public repository and provide us with the relevant URLs, DOIs, or accession numbers. Please see http://www.bmj.com/content/340/bmj.c181.long for guidelines on how to de-identify and prepare clinical data for publication. For a list of acceptable repositories, please see http://journals.plos.org/plosone/s/data-availability#loc-recommended-repositories.

Additional Editor Comments (if provided):

Dear author,

please follow reviewers suggestions to improve the paper.

Reviewers' comments:

Reviewer's Responses to Questions

**Comments to the Author**

1. Is the manuscript technically sound, and do the data support the conclusions?

Reviewer #1: Yes

Reviewer #2: Yes

2. Has the statistical analysis been performed appropriately and rigorously? 

Reviewer #1: Yes

Reviewer #2: Yes

3. Have the authors made all data underlying the findings in their manuscript fully available?

Reviewer #1: Yes

Reviewer #2: Yes

4. Is the manuscript presented in an intelligible fashion and written in standard English?

Reviewer #1: Yes

Reviewer #2: Yes

5. Review Comments to the Author

Reviewer #1: The topic is of interest; The objective is not novel.

The methodology is appropriate and described in detail.

The authors report negative results: no added value of FDG-PET radiomics features in prediction of outcome in advanced NSCLC.

Overall, the study is of good quality. The results could contribute to the current debate on radiomics value Nonetheless, the findings may be dependent on the study conditions, including the patient cohort size, and, therefore, not provide definitive data.

Reviewer #2: This is a very timely paper addressing the robustness of PET radiomic features.

The method is sound and thorough. The paper is well-written and easy to follow.

Thus, only a few very minor comments:

1) As also mentioned by the authors the repeatability testing is not test-retest as is normally expected when discussing repeatability, but a comparison of 2 mid-positions scans generated from the same 4D PET/CT scan. Thus, only very little variance would be expected.

2) The repeatability appears better for MTV2.5 compared to MTV40 - could the authors please comment on this?

3) Typo on p12/235: "Only one? MTV40 was not repeatable"

4) The quality of the figures are not very high - at least when printed.

6. PLOS authors have the option to publish the peer review history of their article (what does this mean?). If published, this will include your full peer review and any attached files.

Reviewer #1: No

Reviewer #2: No

---

## [Author Response · Author response to Decision Letter 0]

19 Jan 2020

For the response to specific reviewer and editor comments, please see the attached file 'response to the reviewers'.

---

## [Editor Report · Decision Letter 1]

24 Jan 2020

Robust, independent and relevant prognostic 18F-fluorodeoxyglucose positron emission tomography radiomics features in non-small cell lung cancer: are there any?

PONE-D-19-30593R1

Dear Dr. Konert,

We are pleased to inform you that your manuscript has been judged scientifically suitable for publication and will be formally accepted for publication once it complies with all outstanding technical requirements.

With kind regards,

Domenico Albano

Academic Editor

PLOS ONE
---

## [Editor Report · Acceptance letter]

11 Feb 2020

PONE-D-19-30593R1 

Robust, independent and relevant prognostic ^18^F-fluorodeoxyglucose positron emission tomography radiomics features in non-small cell lung cancer: are there any? 

Dear Dr. Konert:

I am pleased to inform you that your manuscript has been deemed suitable for publication in PLOS ONE. Congratulations! Your manuscript is now with our production department. 

With kind regards,

on behalf of

Dr. Domenico Albano 

Academic Editor

PLOS ONE